

# Research on melanoma image segmentation by incorporating medical prior knowledge

Hong Zhao, Aolong Wang and Chenpeng Zhang

School of Computer and Communication, Lanzhou University of Technology, Lanzhou, Gansu, China

## ABSTRACT

**Background:** Melanoma image segmentation has important clinical value in the diagnosis and treatment of skin diseases. However, due to the difficulty of obtaining data sets, and the sample imbalance, the quality of melanoma image data sets is low, which reduces the accuracy and the effectiveness of computer aided diagnosis of melanoma image.

**Objective:** In this work, a method of melanoma image segmentation by incorporating medical prior knowledge is proposed to improve the fidelity of melanoma image segmentation.

**Methods:** Anatomical analysis of the melanoma image reveal the star shape of the melanoma image, which can be encoded into the loss function of the UNet model as a prior knowledge.

**Results:** Our experimental results on the ISIC-2017 data set demonstrate that the model by incorporating medical prior knowledge obtain a mIoU (Mean Intersection over Union) of 87.41%, a Dice Similarity Coefficient of 93.49%.

**Conclusion:** Therefore, the model by incorporating medical prior knowledge achieve the first rank in the segmentation task comparing to other models and has high clinical value.

## INTRODUCTION

Melanoma is the most common type of skin cancer and has the highest death rate of all skin cancers. However, melanoma can be cured with a minor surgical operation detected at an early stage. Therefore, segmentation of melanoma images can be used to help dermatologists evaluate and take treatment means as soon as possible.

Initially, basic methods with low segmentation accuracy such as edge detection, threshold segmentation and region segmentation were introduced to segmentation of medical images. With the rapid development of deep learning technology, deep learning models have been widely leveraged in segmentation of skin lesions. However, with the improvement of model accuracy, complex structures and models are increasingly dependent on the number of images and high-quality of them, which is extremely difficult to get. The most common techniques to increase the number of images are data augmentation, such as traditional visual augmentation methods or generative models.

Corresponding author
Aolong Wang,
wang_aolong_0403@163.com

In the context of applying models on low-quality data sets (*Badrinarayanan, Kendall & Cipolla, 2017*; *Ronneberger, Fischer & Brox, 2015*; *Zhou et al., 2018*; *Oktay et al., 2018*; *Alom et al., 2018*; *Li et al., 2018*), the most common techniques are data augmentation, such as traditional visual augmentation methods or generative models (*Izadi et al., 2018*). However, these methods only use the data set itself and do not introduce external information into models. Despite the success of augmentation based techniques, the problems of low-accuracy persist. To solve this problem, medical prior knowledge can be introduced into the segmentation model (*Xie et al., 2021*) to improve the performance of the segmentation map, such as transfer learning, using multi-modal data sets, and incorporating physicians' knowledge. The most effective method is to combine physicians' knowledge, that is, using physicians' medical domain knowledge of doctors for model training before segmentation, during or after segmentation. For example, imitating the training patterns of physicians and prioritizing the diagnosis of more severe samples during training (*Berger et al., 2018*). Or use the general diagnostic mode of doctors (*Wu et al., 2018*).

Since real physicians often do not need a large amount of data to make diagnoses, this method can bypass the problem of reliance on high-quality medical image data sets. However, while the above methods are applied to the segmentation of melanoma images, segmentation results that are obviously inconsistent with the anatomical structure of skin lesions, internal hole errors and external island errors occur in the segmentation map. The use of loss functions that do not encode anatomical priors led to these errors.

To solve this problem, specific shape constraints can be designed according to the speckle shape of the melanoma images and utilized to the model can be trained using that. Since the seminal work of *Veksler (2008)*, the star shape prior has been leveraged in the graph cut algorithm. *Mirikharaji & Hamarneh (2018)* is one of the pioneering works to incorporating star shape priors into segmentation of skin lesion images. They encoded the star shape prior into the loss function to improve the convexity of the segmentation map. However, this method is not powerful enough to penalize the internal error pixels in the prediction map, which will lead to the internal hole errors in the segmentation map. The acuracy of the model is reduced by using this approach.

We aim to proposes a melanoma image segmentation method based on star shape prior, and encode it in loss function as a regularization term. To penalize non-star shape segment, including external and internal errors area. our experimental results demonstrate the improvement of segmentation performance.

## MATERIALS AND METHODS
### Encoding regularization term in the loss function as prior knowledge
#### *Encoding prior regularization term in the loss function*
Comparing with the segmentation of ordinary images, there is a lot of many anatomical prior informations in medical images such as the position, shape and topological structure of organs or lesions. There are several methods and techniques to incorporating anatomical priors of lesions or organs into the model of medical image segmentation. It is one of the methods by learning prior knowledge with generative model and then

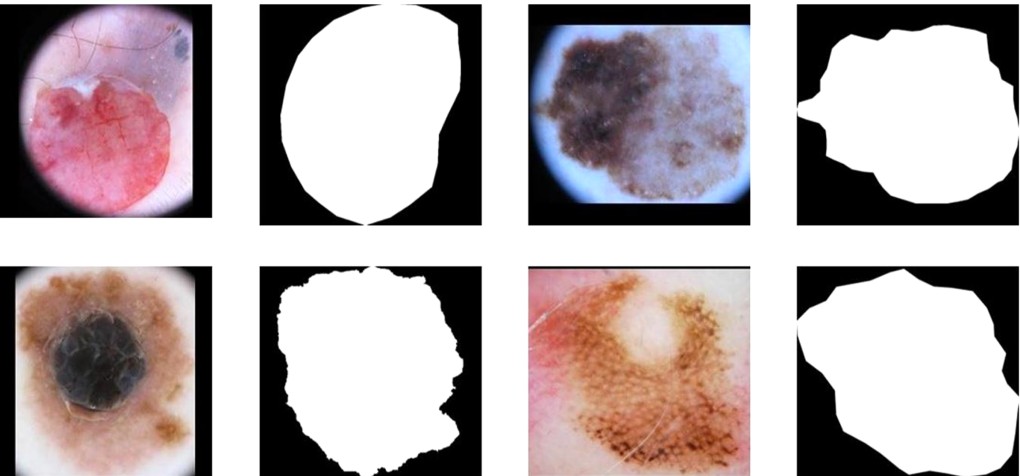

**Figure 1 The internal hole error and external island error in partition diagram.**

integrating it into the network (*Painchaud et al., 2019*). Alternatively, the fuzzy results of segmentation map (*Chen et al., 2019*) can be refined. However, these methods need to be carried out in the pre-processing or post-processing stage and cannot be trained end-to-end.

An end-to-end training model is to encoding anatomic prior knowledge to the loss function as a regularization term. The prior features can be divided into the following categories according to the types of regularization term: topological regularization term, which ignore the shape of organs and extract abstract connections between organs; dimensional regularization terms, which combine dimensional information about the size of organs or lesions in the model; interregional regularization terms are to find the geometrical and distance interaction between image regions; Shape regularization term, such as geometric shape feature, polygon feature, star feature, *etc*.

Different regularization terms should be designed for different organ and lesion when encoding medical prior into the loss function. Topological priors should be used when encoding the features of thin films and curved objects. For the tasks of whole-body segmentation, interregional priors are needed; For skin lesions, such as melanoma segmentation, star priors can be used to improve the convexity of organs or lesions to enhancing the model performance.

Figures 1A, 1C, 1E and 1G demonstrate the original image of melanoma segmentation, and Figs. 1B, 1D, 1F and 1H are the corresponding segmentation labels. As shown in Fig. 1, the color distribution of melanoma lesion sites on the image is asymmetric, leading to degraded segmentation results of the previous model. However, due to malignant melanoma lesion has the anatomical speckled shape, dermatologists often mark the locations of lesions in melanoma images with a star shape. Therefore, it is suitable to encoding star-shaped regularization term to avoid the inner hole error and the outer island error commonly seen in melanoma segmentation models.

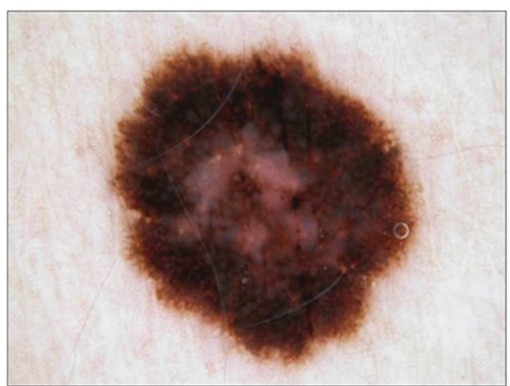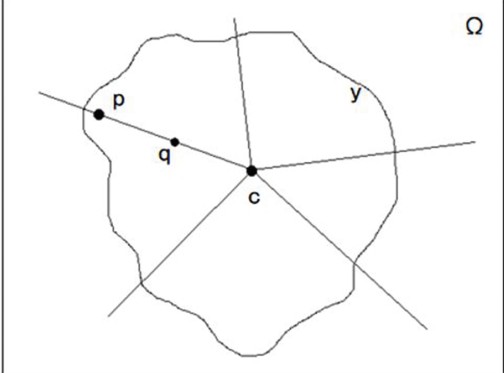

**Figure 2 The original image and segmentation label of melanoma image.**

### Star convex set

Melanoma skin lesions have a speckled shape. The star-shaped prior is suitable for the segmentation of skin lesions. The star-shaped prior is derived from the concept of star convex set in mathematics, as shown in Fig. 2. For a sample of malignant melanoma image, all pixels on the image form pixel space $\Omega$. If all the focal pixels on it form a set $y$, the center of the set is the point $c$. Then, the conditions for identifying $y$ as a star convex set is that: At any point $p$ of the set $y$, its line with $c$ at any point $q$ in $l_{pc}$ is also in $y$.

## Melanoma image segmentation using star prior encoded loss function

There are usually only a few dozens to a few hundreds of malignant melanoma images in a melanoma image data set, which is too small. If only the data set itself is used for image segmentation, the segmentation results will not be able confirm the anatomical characteristics of melanoma skin lesions. Therefore, the UNet model is first constructed in this article. And then segmented star prior loss function was designed for training to enhance the performance of the model.

### Loss function without prior knowledge

There are different type of loss functions without coding anatomic priors, such as cross-entropy loss, Dice loss and Kappa loss, *etc*. The cross-entropy loss function is the most commonly used loss function in segmentation of skin lesions. However, it ignores the quantity ratio of different types of pixels in the sample space. Therefore, it is easy to produce the problem of category imbalance, and the category with the most pixels leads the training inevitably. The Dice loss function can be used to balance positive and negative samples. It has good performance when positive and negative samples are extremely uneven, and the loss of each pixel is strongly correlated with the adjacent region. However, the Dice loss function has the problem of insufficient mining of background area. It will adversely affect the back propagation process, making training process unstable. Therefore, *Zhang, Petitjean & Ainouz (2020)* proposed the Kappa loss function, whose calculation method included all pixels in the image map, even a large number of true negative pixels not involved in other loss functions. The Kappa loss function further

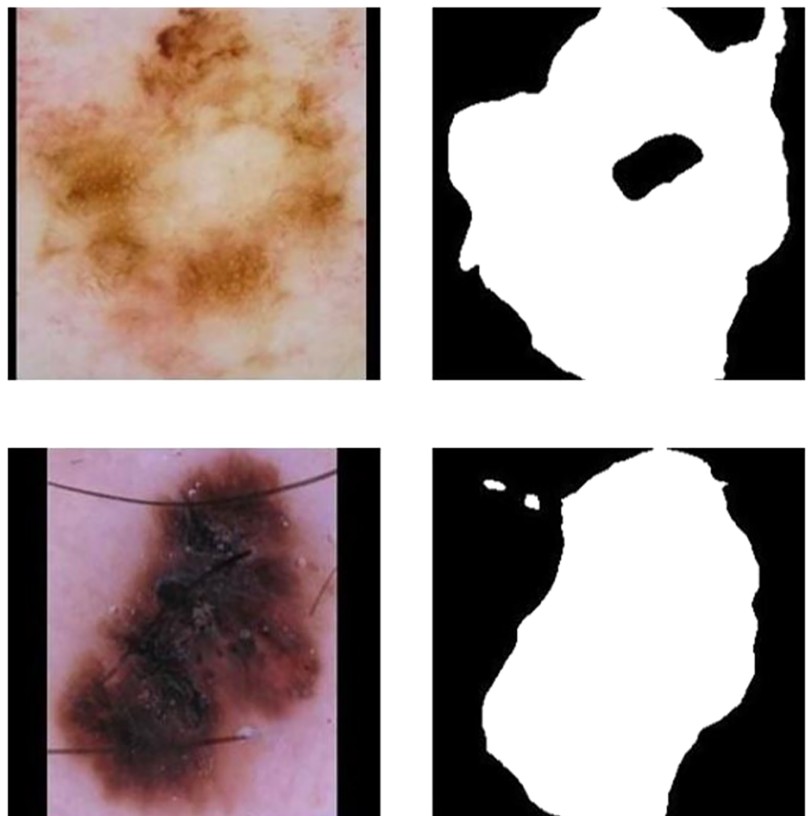

**Figure 3 Schematic of a star convex set in a melanoma image sample.**

improved the segmentation accuracy and model convergence. In addition, new Loss functions such as Focal Loss function (*Lin et al., 2017*) and Conservative Loss (CL) function (*Zhu et al., 2018*) were proposed.

Although these attempts and improvements on loss function have enhanced the segmentation model performance recently, there are still shortcomings as follows (*El Jurdi et al., 2021*): First, these methods ignore the advanced characteristics of lesion location and medical structure. Such as shape and topological structure; Second, they equally penalize all kinds of error pixels in the image segmentation map. While utilizing the above loss functions and using the common segmentation models in segmentation, this model can not take advantage of a particular organ or pathological changes of anatomy structure and the spatial relationship between organs. Therefore, in the segmentation of melanoma images, results that are obviously inconsistent with the anatomical structure of skin lesions. Errors Including internal hole and external island often appear. Figures 3A and 3B show the internal hole error, and Figs. 3C and 3D show the external island error. Although hybrid loss functions have been used recently to obtain high performance (*Goceri, 2021*), we prefered incorporation of prior knowledge to improve efficiency and robustness.

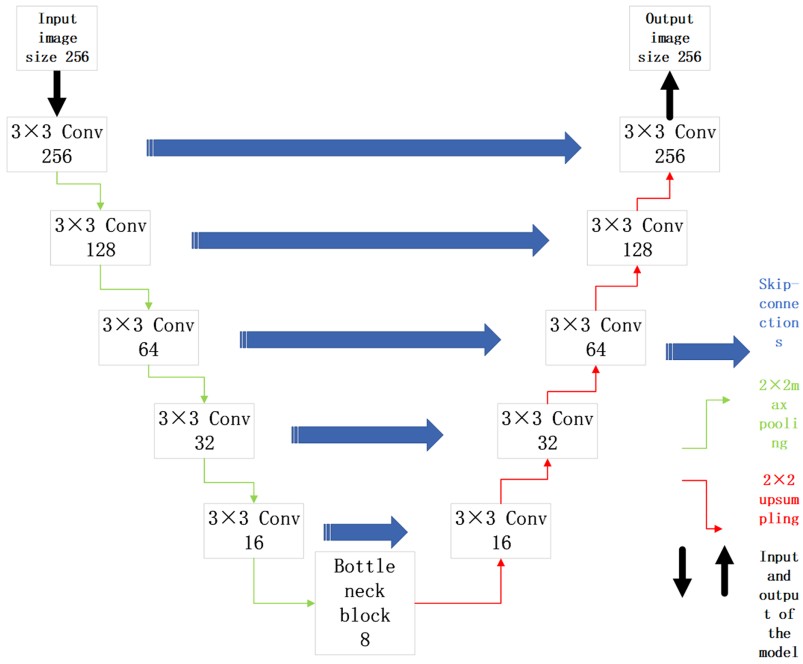

**Figure 4 UNet network model based on ResNet34 network structure.**

### UNet structure

The malignant sample data in the original data set is selected for training. The UNet model is built with ResNet34 as the backbone for image segmentation of malignant samples. Its structure is shown in Fig. 4.

In Fig. 4, the UNet model is divided into an upward path and downward path according to the depth of convolutional layers. The upward path is used to obtain context information, and the downward path is used to locate targets. The high level informations of the network has low resolution but rich semantic information. While The high resolution of low-level informations provide more detailed edge information for segmentation map. Due to such characteristics, the UNet model is suitable for medical image segmentation. Its skip connections structure can not only get high-level semantic informations, but also extract the underlying informations of convolutional network, which conforms to the characteristics of simple semantic information of medical image. The malignant sample data are divided into training set, evaluation set and test set according to a certain proportion. The UNet model based on ResNet34 backbone is built to obtain the location of lesions in the malignant melanoma sample images.

### Star prior loss function applied to segmentation of melanoma

In order to encode the anatomical structure of melanoma skin lesions into the UNet model for training. Shape prior can be fused into the loss function. When the star prior is encoded into the loss function, it can be designed by numerical superposition, as shown in Formula (1) (*Mirikharaji & Hamarneh, 2018*).

$$\theta^* = \arg\min_{\theta} L(X, Y; \theta)$$

(1)

$$L(X, Y; \theta) = \alpha L_{ce} + \beta L_{sh}$$

In Formula (1), $\theta$ represents the UNet parameters, $X$ and $Y$ represents the predicted segmentation map and the segmentation label map respectively. While parameter $\theta$ is continuously optimized until it converges to the final value $\theta^*$. In reference *Berger et al. (2018)*, $L_{ce}$ represents the cross-entropy loss function. It is composed of two parts, $L_{ce}$ and $L_{sh}$, multiplied by weights $\alpha$ and $\beta$ respectively and added together. You can set $\alpha$ to 1 and then adjust the value of $\beta$. A and B are set in the way of hyperparameters before training, where $L_{ce}$ can be replaced by other loss functions commonly used in medical image segmentation, such as Dice loss function.

According to the features of the internal hole errors and external island errors in melanoma image segmentation, the star prior formula regularization term $L_{sh}$ is designed, as shown in Formula (2).

$$L_{sh}(X, Y; \theta) =$$

$$
\begin{cases}
\dfrac{1}{Nn_\Omega} \displaystyle\sum_{i=1}^{N} \sum_{p\in\Omega} \sum_{q\in l_{pc}} \mu * A * B * \dfrac{1}{l_{qc}}, \; y_{ip} = 1 \text{ and } P(y_{ip} = 1|x(i); \theta) < 0.5 \text{ and } PsN(q) < 0 \\[2ex]
\dfrac{1}{Nn_\Omega} \displaystyle\sum_{i=1}^{N} \sum_{p\in\Omega} \sum_{q\in l_{pc}} \rho * A * B * \dfrac{1}{l_{qc}}, \; y_{ip} = 1 \text{ and } P(y_{ip} = 1|x(i); \theta) < 0.5 \text{ and } PsN(q) \geq 0 \\[2ex]
\dfrac{1}{Nn_\Omega} \displaystyle\sum_{i=1}^{N} \sum_{p\in\Omega} \sum_{q\in l_{pc}} A * B * C, \text{ otherwise}
\end{cases}
$$

(2)

$$A = \begin{cases} 1, & \text{if } y_{ip} = y_{iq} \\ 0, & \text{otherwise} \end{cases}$$

$$B = \left| y_{ip} - P(y_{ip} - 1|x(i); \theta) \right|$$

$$C = \left| P(y_{ip} = 1|x(i); \theta) - P(y_{iq} = 1|x(i); \theta) \right|$$

where $N$ is the total number of samples, $\Omega$ is the pixel space of a melanoma image sample, $n_\Omega$ is the total number of pixel points in the pixel space $\Omega$. The location of the focus point is $c$, while $p$ is any point of pixel space $\Omega$, and $q$ is a point on line $l_{pc}$. $y_{ip}$ and $y_{iq}$ represent the label of the points $p$ and $q$, while $y_{ip}$ and $y_{iq}$ represent the model predicted values of points $p$ and $q$ respectively. This symbol $*$ represents matrix multiplication.

$PsN()$ function is used to discriminate whether a point $q$ is an internal hole point. To be specific, it is to connect point $c$ and point $q$ first, and then take the outward ray starting from point $q$ until they touch the edge of the image map. Figure 5A is the segmentation label map of the melanoma image, and the blue part in Fig. 5B represents the prediction map of lesion location by model, where the positions marked in yellow are the segmentation error. The focus of line $l_{qn}$ and internal hole error is $r$. If the length of line $l_{rn}$ is greater than that the length of line $l_{rq}$, point $q$ is identified as the internal hole error point,

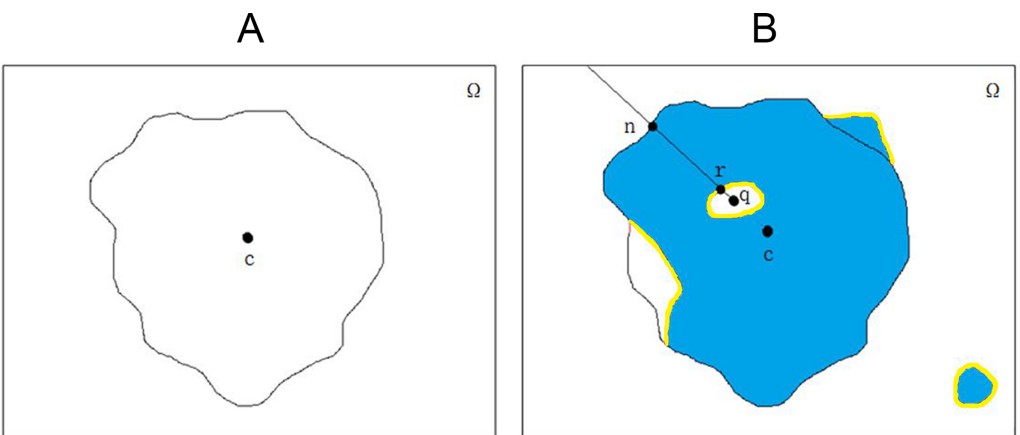

**Figure 5 Determine whether point q is an internal hole.** (A) Segmentation label of melanoma image. (B) Schematic diagram of model prediction.

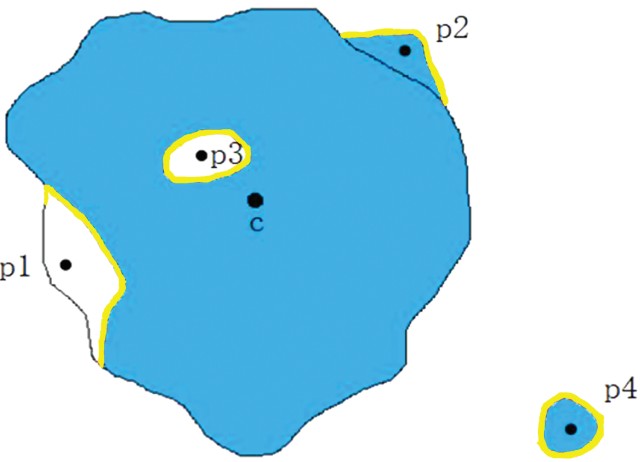

**Figure 6 Different types of segmentation errors.**

namely $\text{PsN}(q) >= 0$; otherwise, point $q$ is identified as the edge false negative error point, namely $\text{PsN}(q) < 0$.

In the medical image segmentation tasks, the errors are divided into two kinds, namely the false positive errors and false negatives errors. Then further subdivided into four categories: the internal false negative, the edge false negative, the external false positive and the edge false positive, as shown in Fig. 6. $p1$, $p2$, $p3$ and $p4$ are classified as edge false negative error, edge false positive error, internal hole error and the external island errors. In melanoma image segmentation, internal false negative (internal hole) errors and external false positive (external island) errors should be penalized. Therefore, $L_{sh}$ was designed as a piecewise function, divided into three parts. In Formula (2), the first formula is designed to regularize the edge false negative errors, the second formula is applied to regularize the internal hole errors, and the third formula is used to normalize the other

outer errors. The hyperparameters $\mu$ and $\rho$ are set separately before training and regularization with different intensities was applied to different kinds of errors.

The optimization goal of the star prior loss function $L_{\text{sh}}$ is to assign the same label to all $q$ points on the line $l_{\text{cp}}$ as to point $p$, on the condition that the labels of point $p$ and point $q$ must be the same. Therefore, item $A$ is designed to be 1 only when the true value of point $p$ and point $q$ are the same. Otherwise, the value will be 0, which means that no shape regular term is encoded. Item $B$ represents the difference between the real value and the predicted values at point $p$, so item $B$ sets the strength of the regularization term for $L_{\text{sh}}$. The more severe the prediction error at point $p$ is compared to the real situation, the larger the regularization term is, and the heavier the penalty for the error is. $A$ and $B$ are divided by $l_{qc}$ when calculating the regularization item. Because the point $q$ and the distance between the center point $c$ will affect the value of the regularization item. If the internal hole error exists segmentation image near the center, the accumulative losses at each point $q$ makes the regularization item too small. Then the loss function cannot achieve the purpose of penalizing these errors; Secondly, the false negative error points near the edge of the real segmentation label are too far away from the center point c, resulting in excessive regularization after loss accumulation. In terms $A$ and $B$, the center distance is scaled by dividing by $l_{qc}$ to remove the influence caused by the distance between point $q$ and point $c$. Instead, $\mu$ and $\rho$ are used to control the value of the regularization term, so as to improve the accuracy.

$C$ sets the way of regularizing the loss function in melanoma lesions. Using the functions as shown in Formulas (1) and (2),which can focus on the penalization of similar error points in Fig. 6 and $p3$ and $p4$. It regularize the final shape of the segmentation map into a star shape, in accordance with the anatomical structure of the melanoma images.

## RESULTS
### Experimental design
#### Data set
The Society for Medical Impact Informatics (SIIM) has collaborated with the International Skin Imaging Collaboration (*ISIC Archive, 2020*), has constructed the largest open library of images of skin lesions. The ISIC Melanoma Data set (*Painchaud et al., 2019*) is referred to as ISICM. The ISICM contains 10,000 dermatoscopic images of skin lesions totally. Each of which is accompanied by a professionally described and segmented image. ISICM is divided into benign and malignant categories according to the disease description, and only 220 images of malignant melanoma were concluded in this data set.

In our work, the model is applied to the preprocessed ISICM data set to build a three-stage processing structure. And the ISICM was finely segmented. Finally, our experiment is demonstrated in images, then the performance of the model is analyzed. 220 malignant sample images were segmented in the proportion of 6:2:2 to the training set, the evaluation set and the test set.

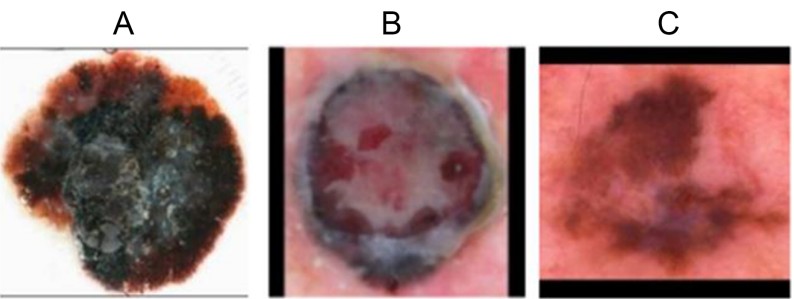

**Figure 7 Malignant samples after preprocessing.** (A) Sample 0155 after preprocessing. (B) Sample 0144 after preprocessing. (C) Sample 0169 after preprocessing.

### Preprocessing

First, the images is scaled to a size of 256 × X or Y × 256 in accordance with the length and width ratio of the image (X and Y are the height and weight of the scaled image. Scale final height and weight in the same proportion as the original image). The edges are then filled and the image size is 256 × 256. In addition, the image is flipped and brightness adjusted. Three malignant samples after data pretreatment are shown in Fig. 7.

### Implementation details

The Pytorch framework (*Zhu et al., 2018*) was used to implement this method (*Paszke et al., 2019*). Adam was selected as the optimizer during the training process. The initial learning rate was set in 1E−3. In our experiment, 400 epochs were trained. ImageNet pre-trained model (*Russakovsky et al., 2015*) was used for freezing training for the first 100 epochs, then learning rate attenuation was applied for the last 200 epochs during training (*Loshchilov & Hutter, 2017*), in which every five epochs were attenuated to 95% of the original learning rate. When the model is tested, the output of the model is taken as the final result without any subsequent processing. Our model was run on a NVIDIA GTX1060.

## Ablation experiments

For the core experiment, three groups of ablation experiments were designed, namely, the non-regular term group, the unsegmented star prior group and the non-central distance scaling group. The experimental results are shown in Table 1.

The segmentation results are displayed, as shown in Fig. 8.

### Non regularization term group (baseline)

As the baseline experimental group, the loss function with star priori was not used, and only Dice loss function was used for training. As shown in Fig. 8C, due to the absence of shape regularization, the segmentation results do not conform to the anatomical structure of melanoma, resulting in poor accuracy.

### Unsegmented star prior group

Unsegmented basic star prior formula. Item $L_{sh}$ design is shown in Formula (3).

**Table 1 The experimental results.**

| The experimental group | Choice of $L_{ce}$ function | Parameter settings | | | | mIoU (%) | mAP (%) | Dice (%) | Jaccard (%) |
|---|---|---|---|---|---|---|---|---|---|
| | | Item $L_{ce}$ weight $\alpha$ | Item $L_{sh}$ weight $\beta$ | item $A$ weight $\mu$ | item $B$ weight $\rho$ | | | | |
| Non-regular item (baseline) group | Cross entropy loss | None | None | None | None | 85.10 | 92.56 | 92.68 | 86.76 |
| | Dice loss | None | None | None | None | 85.72 | 92.61 | 92.89 | 86.95 |
| Unsegmented star prior group | Cross entropy loss | 1 | 0.01 | None | None | 85.31 | 92.61 | 92.86 | 86.98 |
| | Dice loss | 1 | 0.01 | None | None | 85.90 | 92.65 | 93.10 | 87.18 |
| Non-center distance scaling group | Cross entropy loss | 1 | 0.33 | 0.02 | 0.05 | 84.97 | 92.57 | 92.79 | 86.78 |
| | | | | 0.02 | 0.1 | 84.79 | 92.50 | 92.71 | 86.65 |
| | Dice loss | 1 | 0.33 | 0.02 | 0.05 | 85.42 | 92.61 | 92.97 | 87.01 |
| | | | | 0.02 | 0.1 | 85.38 | 92.56 | 92.92 | 86.85 |
| Segmented star prior group | Cross entropy loss | 1 | 0.33 | 0.1 | 0.5 | 86.48 | 92.89 | 93.05 | 87.22 |
| | | | | 0.1 | 1 | 86.78 | 93.08 | 93.22 | 87.50 |
| | Dice loss | 1 | 0.33 | 0.1 | 0.5 | 87.09 | 92.95 | 93.26 | 87.40 |
| | | | | 0.1 | 1 | 87.41 | 93.13 | 93.49 | 87.69 |

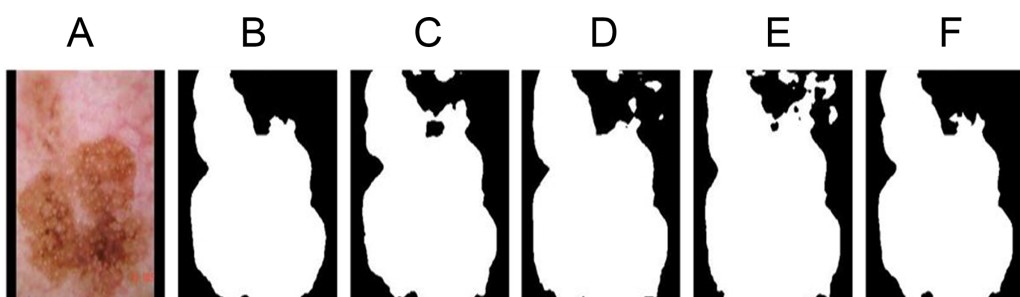

**Figure 8 Ablation experiment segmentation results display.** (A) The original image 1138. (B) Segmentation label. (C) Non-regular item group. (D) Unsegmented star prior group. (E) Non-center distance scaling group. (F) Segmented star prior group.

$$L_{sh}(X, Y; \theta) = \sum_{i=1}^{N} \sum_{p \in \Omega} \sum_{q \in l_{pc}} B^i_{pq} \times |y_{ip} - P(y_{ip} = 1|X(i); \theta)|$$
$$\times |P(y_{ip} = 1|X(i); \theta) - P(y_{iq} = 1|X(i); \theta)|;$$
$$B^i_{pq} = \begin{cases} 1, if\, y_{ip} = y_{iq} \\ 0, otherwise \end{cases} \tag{3}$$

As shown in Fig. 8D, the unsegmented star prior term uses the same regularization term to penalize all errors that do not conform to the anatomical structure of the lesions, leading to the difficulty in selecting weights $\alpha$ and $\beta$ when designing the $L_{sh}$ term, and to penalize the segmentation errors of internal hole and external island with appropriate size. Because

**Table 2  Results of different methods.**

| Model | mIoU (%) | mAP (%) | Dice (%) | Jaccard (%) |
|---|---|---|---|---|
| UNet | 86.00 | 92.53 | 92.97 | 87.09 |
| FC-DenseNet | 85.92 | 92.66 | 93.04 | 87.12 |
| SSPFCN | 85.90 | 92.65 | 93.10 | 87.18 |
| Attention UNet | 86.56 | 92.76 | 93.16 | 87.35 |
| Our method | 87.41 | 93.13 | 93.49 | 87.69 |

the loss function does not use the segmentation function for various kinds of errors, the segmentation result is reduced.

### Non-center distance scaling group

The segmented star prior loss function in Formula (1) and Formula (2) is applied for training, but $l_{qc}$ is no division at the end of items $A$ and item $B$. Only the parameters $\mu$ and $\rho$ are used to control the value of the regularization term. As shown in Fig. 8E, because each point $q$ in the segmentation map and the distance between the center point $c$ each are not identical. That will have an effect on the value of the regularization item, so using no $L_{sh}$ center distance scaling way designs, $\mu$ and $\rho$, regardless of what parameters the model select will cause a dramatic performance results decline. The performance is even lower than the non regularization terms group.

### Segmented star prior group

The segmented star prior loss function in Formulas (1) and (2) was used for training, and the results were shown in Fig. 8F. The segmentation results were consistent with the anatomical results of melanoma with high accuracy. Using this improved loss function, the convexity of the shape can be enhanced.

## Compare with other segmentation models

Compared with UNet fused star prior method SSPFCN (*Li et al., 2018*), Attention UNet (*Oktay et al., 2018*) and dense convolution method FC-densenet (*Jégou et al., 2017*) on the ISICM data set, the results are shown in Table 2.

Previous methods performed well on large, high-quality data sets, but performed poorly on ISIC data set with only 220 skin images of malignant melanoma. Comparing to the recent methods, the star priority-based melanoma segmentation method has further improved the performance of the segmentation model on the data set with fewer samples. An advantage of the proposed approach is that a separate intensity normalization stage (*Goceri, 2018*), which usually leads to increase the computational complexity, is not needed.

## CONCLUSIONS

In this work, a melanoma segmentation method incorporating medical prior is proposed, and experimental verification is carried out on ISICM data set. Piecewise loss function of star prior is designed and applied to melanoma image segmentation teak. Our experiment

demonstrate that the melanoma segmentation model based on the star prior enhances the performance of the model, and improves the mIoU value, mAP value, Dice value and Jaccard value.

### Funding
This work was supported by the National Natural Science Foundation of China (No. 62166025) and the Key R & D program of Gansu Province (No. 21YF5GA073). The funders had no role in study design, data collection and analysis, decision to publish, or preparation of the manuscript.

### Grant Disclosures
The following grant information was disclosed by the authors:
National Natural Science Foundation of China: 62166025.
Key R & D program of Gansu Province: 21YF5GA073.

### Competing Interests
The authors declare that they have no competing interests.

### Author Contributions
- Hong Zhao conceived and designed the experiments, analyzed the data, authored or reviewed drafts of the article, and approved the final draft.
- Aolong Wang conceived and designed the experiments, performed the experiments, analyzed the data, performed the computation work, prepared figures and/or tables, and approved the final draft.
- Chenpeng Zhang conceived and designed the experiments, performed the computation work, prepared figures and/or tables, authored or reviewed drafts of the article, and approved the final draft.

### Data Availability
The code and data are available in the Supplemental Files.

### Supplemental Information
Supplemental information for this article can be found online at http://dx.doi.org/10.7717/peerj-cs.1122#supplemental-information.

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
