# Peer review of "Research on melanoma image segmentation by incorporating medical prior knowledge"

_PeerJ Computer Science, doi:10.7717/peerj-cs.1122_

## Round 0.1 · original submission · Major Revisions

In the view of the reviewers, I would like the authors to revise their manuscript and carefully address all the raised concerns. I would also suggest authors polish the reference list and include the recent studies in the literature as the reference list is relatively small. In addition, please polish the formatting of the manuscript and improve the quality of figures, symbols, equations, and tables. Also, carefully proofreading the manuscript against grammatical mistakes and typos as already pointed out by reviewers.

Finally, Reviewer 1 has requested that you cite specific references. You may add them if you believe they are especially relevant. However, I do not expect you to include these citations, and if you do not include them, this will not influence my decision.

Reviewer 1 ·

Basic reporting

The paper should be re-read and updated to correct grammatical mistakes.
For instance, there are grammatical mistakes in these sentences,
"The loss functions without coded anatomic priors knowledge including cross-entropy loss,Dice loss and Kappa loss,etc."
and
"While utilizing the above loss functions and the common models in segmentation,such as UNet model."

Experimental design

Rigorous investigation has been performed.

Validity of the findings

All underlying data have been provided.

Additional comments

In this paper, a method of melanoma image segmentation by incorporating medical prior knowledge is proposed.

Skin lesion segmentation, particularly melanoma segmentation, is an important topic and active research area.
So, the subject is worth investigating.
However, the following revisions are required to improve the quality of the paper;

1)
In the introduction section, the sentence,
"With the rapid development of deep learning technology, deep learning model has been widely leveraged in medical image segmentation"
is about medical image segmentation (quite general). However, the topic handled in this work is about melanoma segmentation, in other words, skin lesion segmentation.
Also, the meaning of the sentence should be supported with appropriate works published recently.
Therefore, the sentence should be updated as follows;
"With the rapid development of deep learning technology, deep learning models have been widely leveraged in segmentation of skin lesions [R1-R5]."
R1:"Automated Skin Cancer Detection: Where We Are and The Way to The Future", https://ieeexplore.ieee.org/abstract/document/9522605, doi: 10.1109/TSP52935.2021.9522605, 2021
R2:"Capsule Neural Networks in Classification of Skin Lesions", The 15th Int.Conf. on Computer Graphics, Visualization, Computer Vision and Image Processing (CVGCVIP 2021)
R3:"Skin disease diagnosis from photographs using deep learning", https://doi.org/10.1007/978-3-030-32040-9_25
R4:"Convolutional neural network based desktop applications to classify dermatological diseases", https://ieeexplore.ieee.org/abstract/document/9334956, doi: 10.1109/IPAS50080.2020.9334956.
R5:"Impact of deep learning and smartphone technologies in dermatology: Automated diagnosis", https://ieeexplore.ieee.org/abstract/document/9286706, doi: 10.1109/IPTA50016.2020.9286706

2)
In the introduction section, these sentences,
"Medical image segmentation models based on deep learning.......improve the segmentation accuracy of the model."
are redundant/unnecessary, so they should be removed.
Similarly, the following sentences should be removed:
"While determining nodules in CT images, radiologists often change the width and the center of the window [10]."
"Although the above medical image segmentation methods have ......"

3)
These sentences,
"However,with the improvement of model accuracy,complex structures and models are increasingly dependent on high-quality data sets, which is extremely difficult to get. In the context of applying models on low-quality data sets,the most common techniques are data augmentation, such as traditional visual augmentation methods or generative models [7]."
should be updated since image augmentations are applied to increase the number of images rather than image qualities, and also
one of the factors affecting accuracies are the number of images (not only quality of images).
To overcome the low-quality problem, transfer learning or integration of prior knowledge are the most common approaches.

Also, the reference given in [7] was published in 2018 (there are many works published more recently).
In addition, the reference is not appropriate for the above sentence since a reference on image augmentations should be presented for the sentence.
So, I suggest updating the sentence as:
"However,with the improvement of model accuracy, complex structures and models are increasingly dependent on the number of images and high-quality of them, which is extremely difficult to get. The most common techniques to increase the number of images are data augmentation, such as traditional visual augmentation methods or generative models [R1,R2]."
R1:Image augmentation for deep learning based lesion classification from skin images, https://ieeexplore.ieee.org/abstract/document/9334937, doi: 10.1109/IPAS50080.2020.9334937
R2:Challenges and recent solutions for image segmentation in the era of deep learning, https://ieeexplore.ieee.org/abstract/document/8936087, doi: 10.1109/IPTA.2019.8936087

4)
The section named as "Loss function without prior knowledge" should be out of the section "Materials & Methods", which should present the proposed method and materials used in the work

5)
The sentence, "The cross-entropy loss function is the most commonly used loss function in medical image segmentation" should be updated
by saying "skin lesion segmentation" instead of "medical image segmentation" also the meaning of the statement should be supported with references.
So, the sentence should be updated as:
"The cross-entropy loss function is the most commonly used loss function in segmentation of skin lesions [ ]"
R1:"Comparative evaluations of cnn based networks for skin lesion classification", Int.Conf. on Computer Graphics, Vis., Computer Vision and Image Processing,https://www.cgv-conf.org/wp-content/uploads/2020/07/03_202011C031_S044.pdf
R2:"Analysis of deep networks with residual blocks and different activation functions: classification of skin diseases",https://ieeexplore.ieee.org/abstract/document/8936083

6)
Recently, hybrid loss functions have been used in deep networks to obtain high performance. However, in the section named "Loss function without prior knowledge"
hybrid loss functions have been ignored. To inform the readers about them, at the end of the section, after the sentence,
".......and Figures 1(c) and (d) show the external island error."
the following statement should be added:
"Although hybrid loss functions have been used recently to obtain high performance [R1-R3], we prefered incorporation of prior knowledge to improve efficiency and robustness."
R1:"Deep learning based classification of facial dermatological disorders", https://doi.org/10.1016/j.compbiomed.2020.104118
R2:"Diagnosis of skin diseases in the era of deep learning and mobile technology", https://doi.org/10.1016/j.compbiomed.2021.104458
R3:"An Application for Automated Diagnosis of Facial Dermatological Diseases", https://dergipark.org.tr/en/pub/ikcusbfd/issue/65176/848630

7)
Limitation/drawbacks/disadvantages of the proposed approach should be explained in the last section.
Also, future extensions of the approach can be presented.
In addition, an important advantage of the proposed approach is that it is not necessary to implement a separate intensity normalization stage, which usually leads to increase the computational complexity.
Therefore, the following statement should be added in the last section to inform the readers about the property of the proposed approach;
"An advantage of the proposed approach is that a separate intensity normalization stage (e.g., [R1,R2]), which usually leads to increase the computational complexity, is not needed."
R1: Fully Automated and Adaptive Intensity Normalization Using Statistical Features for Brain MR Images, https://doi.org/10.18466/cbayarfbe.384729
R2: Intensity Normalization in Brain MR Images Using Spatially Varying Distribution Matching, 11th Int. Conf. on Computer Graphics, Visualization, Computer Vision and Image Processing (CGVCVIP), Lisbon, Portogual

Reviewer 2 ·

Basic reporting

- There are so many bad-formatting e.g.
x No space after full stop "."
x No align or justify texts
x Having extra space e.g. lines 80-88

- Bot compete sentences e.g. lines92-93

- In the equation (2), all symbols must be well defined and the * operation must be defined.

Experimental design

- In the equation (1), alpha and beta are introduced to weight the two losses. However, the authors must explain how to select the values for these two parameters. Are they normalized ? At least the authors should demonstrate the experimental results of attempting multiple combinations of the values of these two parameters.

- In line 255, what are X and Y?
- In line 258, what are 0155, 0144, 0169 ?

- The results are only slight better than the baselines. The authors may need to emphasize the advantage of the proposed loss function by performing additional experiments on (1) more datasets (2) other types of cancer/organs.

- In table 1, why only fixed valued of alpha and beta were shown? How to optimize the values?

Validity of the findings

- The technical contribution is limited. The proposed method is based on the well-known UNet with the revised loss function. However, the results were slightly better than the baseline methods. The authors must find the way to emphasize this contribution e.g. do a more comprehensive analysis and discussion of the performances and segmentation results (segmented output images) to demonstrate the benefit of having this new loss function

Additional comments

- The methods sub-part of the abstract was not well written. It missed out details summarization of the key contribution of the proposed method.

- The authors must provide the evidences e.g. experiments or refs to proof the claims in line 74-76.

- The conclusion was not well written. The key finding, contributions, future works, and generalization of the proposed method should be stated.

---

## Round 0.2 · accepted · Accept

I think the manuscript is ready for acceptance. However, I invite the authors to proofread the manuscript against any grammatical and typos.

Reviewer 1 ·

Basic reporting

The paper has been revised carefully.
Formal results include clear definitions of all terms and theorems and detailed proofs.

Experimental design

Good. Methods are described with sufficient detail.

Validity of the findings

All underlying data have been provided;

Additional comments

It can be accepted